# Transcriptome Analysis of Cisplatin, Cannabidiol, and Intermittent Serum Starvation Alone and in Various Combinations on Colorectal Cancer Cells

**DOI:** 10.3390/ijms241914743

**Published:** 2023-09-29

**Authors:** Viktoriia Cherkasova, Yaroslav Ilnytskyy, Olga Kovalchuk, Igor Kovalchuk

**Affiliations:** Department of Biological Sciences, University of Lethbridge, Lethbridge, AB T1K 3M4, Canada; viktoriia.cherkasova@uleth.ca (V.C.); slava.ilyntskyy@uleth.ca (Y.I.)

**Keywords:** cisplatin, cannabidiol, intermittent fasting, serum starvation, colorectal cancer

## Abstract

Platinum-derived chemotherapy medications are often combined with other conventional therapies for treating different tumors, including colorectal cancer. However, the development of drug resistance and multiple adverse effects remain common in clinical settings. Thus, there is a necessity to find novel treatments and drug combinations that could effectively target colorectal cancer cells and lower the probability of disease relapse. To find potential synergistic interaction, we designed multiple different combinations between cisplatin, cannabidiol, and intermittent serum starvation on colorectal cancer cell lines. Based on the cell viability assay, we found that combinations between cannabidiol and intermittent serum starvation, cisplatin and intermittent serum starvation, as well as cisplatin, cannabidiol, and intermittent serum starvation can work in a synergistic fashion on different colorectal cancer cell lines. Furthermore, we analyzed differentially expressed genes and affected pathways in colorectal cancer cell lines to understand further the potential molecular mechanisms behind the treatments and their interactions. We found that synergistic interaction between cannabidiol and intermittent serum starvation can be related to changes in the transcription of genes responsible for cell metabolism and cancer’s stress pathways. Moreover, when we added cisplatin to the treatments, there was a strong enrichment of genes taking part in G2/M cell cycle arrest and apoptosis.

## 1. Introduction

According to the World Health Organization, colorectal cancer (CRC) is the third most common cancer and the second leading cause of cancer-related deaths worldwide. It is often diagnosed at later stages when the treatment options are limited [1].

CRC is not a single disease but rather a genetically heterogeneous group of oncological pathologies [2]. In 2014, four consensus molecular subtypes (CMSs) of CRC were developed based on genetic alterations affecting the colonic epithelium: CMS1—microsatellite instability (MSI)-immune (CpG island methylation phenotype, and BRAF mutation associated with immune cell infiltrates); CMS2—canonical (WNT- and MAPK-associated); CMS3—metabolic (KRAS-associated); and CMS4—mesenchymal [3,4]. Therefore, there is no “best” single drug for CRC treatment, and different therapeutic approaches should be undertaken based on the stage, subtype, and pathogenesis of CRC [2,5,6].

Despite the implementation of various preventive strategies, screening protocols, and a diverse range of treatment choices, CRC remains a prominent contributor to global disease prevalence and mortality. There is a pressing demand for the development of new, more efficient prevention methods and therapeutic strategies that involve drug combinations. According to recent experimental findings, cannabinoids might emerge as promising contenders in this regard [7,8,9,10,11,12,13,14,15].

Over recent years, multiple experimental data have provided evidence of the antioncogenic impact of cannabinoids on CRC [16,17,18,19,20]. The mechanisms underlying the anticancer properties of cannabinoids, like cannabidiol (CBD), involve triggering apoptosis, activating the endoplasmic reticulum (ER) stress response, reducing survivin (an apoptosis inhibitor), and diminishing RAS/MAPK and PI3K/AKT signaling pathways [7,11,16,19,21]. 

CBD is a partial agonist of cannabinoid receptors 1 and 2 (CB1 and CB2) [22,23]. Additionally, CBD activates transient ion receptor ion channels (TRPV1, TRVP2) [24,25,26,27,28,29], peroxisome proliferating activated receptor α (PPAR α), and PPAR γ, inhibits GPR55 [28,29,30], and increases endocannabinoid anandamide (AEA) concentration by blocking its hydrolysis [31]. Many of the listed receptors are expressed in the gastrointestinal tract [32,33,34,35,36]. Although many receptors are involved in cannabinoid effects on cancers, we did not prioritize the action of specific receptors on tested colorectal cancer cell lines. Our work mainly focused on the drug combination effects on colorectal cancer cell lines with different genetic landscapes.

One of the best-described anticancer effects of CBD is the activation of NOXA, suppressing mTOR/AKT signaling and MAPK pathways [20]. In CRCs, the RAS-MAPK pathway is overactivated, with KRAS and BRAF being overexpressed in approximately 50% and 15% of cases, respectively [37,38]. PI3K/AKT signaling is upregulated in almost 40% of colon malignancies [39]. 

In a study involving the combination of CBD with the platinum drug oxaliplatin, it was shown that oxaliplatin resistance can be overcome by adding CBD to the treatment, which results in autophagy-mediated cell death and the formation of free radicals by dysfunctional mitochondria in resistant cells [40]. Furthermore, other studies regarding drug resistance and cannabinoids indicated that THC, CBD, and CBN could inhibit ATP-binding cassette family transporters, P-glycoprotein, and the breast cancer resistance protein (BCRP) [41], resulting in the potential chemosensitizing effect of cannabinoids in resistant CRCs [42,43,44]. These data were one of the reasons why we decided to combine cisplatin, a commonly used chemotherapeutic agent, with CBD.

A DNA crosslinking agent, cisplatin, is used in multiple combination therapies with paclitaxel, tegafur-uracil, doxorubicin, gemcitabine, and vitamin D [45]. Unfortunately, any chemotherapy, including cisplatin, has a narrow therapeutic window and multiple adverse effects [46,47]. The ability to generate DNA crosslinks culminates in activating cell cycle checkpoints. Cisplatin temporarily induces S phase arrest facilitated by p16. G2/M cell cycle arrest is more prominent due to potent inhibition of Cdc2-cyclin A or B kinase. Cisplatin also activates ATM and ATR, which causes the phosphorylation of the p53 protein. Cisplatin-induced ATR activation results in the upregulation of CHK1 kinase and directly CHK2, which is independent of ATM. One of the major cascades activated by ATR is MAPK. The MAPK signaling cascade includes ERK, JNK, SAPK, and p38, which regulate cell proliferation, differentiation, cell survival, and apoptosis [48], with ERK being one of the most prominent inducers of apoptosis [49].

One of the crucial pathways of cisplatin-induced apoptotic death is the activation of p38 MAPK, which causes transcription of PUMA and NOXA through p53 activation [50,51]. Hayakawa et al. (2000) [52] demonstrated that cisplatin-induced DNA damage leads to the phosphorylation of BAD via the pro-survival AKT pathway. More recent studies have also shown that BAD protein was phosphorylated with the help of the ERK cascade under cisplatin treatment [53]. Inhibition of AKT or ERK cascades caused sensitization of ovarian cancer cells to cisplatin [52]. These mechanisms are also commonly affected by cannabinoids, which provides evidence for the combination of cisplatin and CBD as holding the possibility of synergistic interaction in CRC cell lines.

Humanity has been practicing fasting for centuries. However, only recent years’ discoveries have shown the health benefits of simple time-adapted food consumption and calorie deprivation. Fasting helps reprogram cellular stress-related adaptation mechanisms and energy metabolism and boosts cellular defense mechanisms. It was shown in animal models that intermittent fasting (IF) could protect against diabetes mellitus, cardiovascular pathologies, neurodegeneration, and cancer. Recent human trials showed that IF helps fight obesity, hypertension, asthma, and rheumatoid arthritis. Fasting may lengthen the lifespan of bacteria, yeast, worms, and mice [54,55,56,57]. 

Research showed that serum starvation in vitro and short-term food starvation in vivo reduced levels of growth factor stimulation [58,59,60]. In normal cells, the depletion of growth signals decreases the activity of proliferation-stimulating signaling and reduces metabolism [61]. However, in cancer cells, starvation increases cellular stress due to their metabolism reprogramming to maintain continuous proliferation [62] and activates the DNA damage response [63]. 

Another animal study including fasting showed that alternate-day fasting in mice reduced the incidence of lymphomas [64]. The same study also showed that the significant reduction in the formation of reactive oxygen species (ROS) in mitochondria was associated with the upregulation of superoxide dismutase (SOD) activity [64]. Fasting one day a week could delay spontaneous tumorigenesis in p53-deficient mice [65]. The mechanism of cancer delay was p53-independent and IGF-1-related. These studies identified possible human energy balance interventions to prevent cancer development [65]. However, refeeding may cause abnormally high cell proliferation [66]. In some mice models, periodic fasting can be as effective as chemotherapy [67]. Periodic fasting may also sensitize some tumors to chemotherapy medications because, in contrast to normal cells, cancer cells cannot adapt to fasting due to the accumulation of mutations responsible for cell growth enhancement in normal environmental conditions [67]. 

In metastatic animal models, the combination of fasting and chemotherapy resulted in 20–60% cancer-free survival compared to chemotherapy or fasting alone [63,67]. Additionally, studies on neuronal protection showed that IF might boost antioxidant defense and heat shock proteins and decrease levels of pro-inflammatory factors such as TNF-α, IL-1β, and IL-6 [68]. Thus, inhibition of the mTOR pathway, activation of autophagy, and ketogenesis also benefit the human body, which would help fight cancer [69,70]. 

The inhibition of the glycolytic pathway in cancer cells could lead to stress overload due to constant proliferative signaling from oncogenes [71]. Cancer frequently encounters hypoxia-reperfusion in its microenvironment, leading to extreme ROS production that causes mitochondrial damage and apoptosis [72]. Thus, tumors learn to alleviate oxidative damage via increased glycolysis and downregulate mitochondrial function [72]. The therapeutic perspective of tumor metabolic modulation has yet to be extensively studied. There are a few ways to influence tumor cell metabolism that could have therapeutic implications and normalize cell metabolism. First is inhibition of HIF-1α, which can inhibit angiogenesis, next is the re-establishment of p53, which can activate apoptosis, and last is suppression of the PI3K/AKT/mTOR signaling pathway, inhibiting cell growth and proliferation [73]. Thus, in glucose-addicted cancer cells that have overactivated oncogenes, low-nutrient conditions and inhibitors of glycolysis may enhance the activation of apoptosis.

The mentioned metabolic changes in cancer cells might become one of the targets for anticancer therapy. As we discussed before, fasting affects metabolism in cancer cells, leading to differential stress sensitization. Additionally, cannabinoids are also known to regulate stress survival pathways in cancer cells. Thus, it makes sense to test these treatments individually and in combination, to assess if there is a cytotoxic effect on CRC cells and whether these treatments can act in synergy.

## 2. Results

### 2.1. A Combination of ISS with CBD, ISS with Cisplatin, or ISS and CBD with Cisplatin Showed Synergistic Interaction

First, to evaluate the treatment effectiveness, we established the time and dose-dependent effects (Figure 1) and calculated the IC50 concentrations (Table 1) for all tested treatments in three CRC cell lines. Results were based on an MTT assay.

Next, we combined various treatments in CRC cell lines to test if any combination had a synergistic interaction (Figure 2). Actual CI values for different drug combinations in the HCT-116 cell line are shown in Appendix A. Cell viability results showed a strong synergistic effect between ISS and CBD in all tested cell lines. Surprisingly, the combination of CBD with cisplatin showed antagonistic interaction. However, when we combined the three treatments together, i.e., ISS, CBD, and cisplatin, the interaction effect showed synergy. Additionally, combining cisplatin with ISS had various results, with a moderate antagonistic/additive effect in the HCT-116 CRC cell line and synergism in the HT-29 and the LS-174T cell lines. To acquire some understanding of the molecular events behind the drug effects alone and their interactions, we performed mRNA expression analysis of the HCT-116 CRC cell line. This cell line was used because the cisplatin IC50 level was the lowest compared to the other cell lines, and the use of a lower amount of cisplatin would allow for less “noise” at the level of mRNA in the treated cells.

### 2.2. Cisplatin Changed the Transcription of Genes Responsible for p53-Mediated Cell Death in the HCT-116 CRC Cell Line

Based on the differential expression analysis in the HCT-116 CRC cells, cisplatin increased the expression of genes responsible for p53-mediated cell cycle arrest in G1/S and G2 phases, activation of the DNA-damage response, including *SNF*, and apoptosis, such as *BAX* and *FAS*. These results were somewhat expected based on the drug mechanisms of action (Figure 3A,B). Additionally, the Reactome analysis showed an enrichment of the syndecan interactions and the extracellular matrix organization terms.

One of the most enriched terms under cisplatin action on the HCT-116 CRC cell line was TP53-regulated transcription of the cell death gene (Table 2). The increased levels of *TP53*, *BAX*, *FAS,* and *TNF* receptor ligands indicate that cells were heading toward apoptosis. The increased transcript levels of cyclin-dependent kinase inhibitor 1A (*CDKN1A*), which binds to cyclin/cyclin-dependent kinase 2 or 4 complexes and functions as a regulator of the G1 cell cycle, could support the notion of G1/S cell cycle arrest. Increased *RGCC* gene expression is also responsible for cell cycle regulation and is induced by p53 under DNA damage. Additionally, higher levels of *BTG* anti-proliferation factor 2 could have helped with G1/S cell cycle arrest. The increased levels of stratifin (*SFN*) transcript, whose product binds to translation initiation factors and functions as a regulator of translation during mitosis, help prevent DNA errors during mitosis and could indicate cancer cell survival. The increased transcription of the *SESN1* gene, coding for sestrin 1, as induced by p53 and serving as a potent inhibitor of mTOR, would help in cisplatin’s cytotoxic effects. Although the increased levels of polo-like kinase 2 might also indicate a strong stimulus for cell proliferation, the weight of anti-proliferative signaling was stronger, which supports our MTT results. The increased levels of fibroblast growth factor 2 (*FGF2*) and syndecan 1 (*SDC1*), responsible for cell binding, cytoskeletal organization, interactions with the extracellular matrix (ECM), and cell migration, could stimulate epithelial-to-mesenchymal transition (EMT) and invasiveness of the HCT-116 cell line. These are the signs of cancer cell progression.

### 2.3. CBD Changed the Transcription of Genes Responsible for Carbohydrate Metabolism in the HCT-116 CRC Cell Line

Based on the dot plot data, CBD significantly affected carbohydrate metabolism, PPAR, receptor-type tyrosine-protein phosphatase, TGF-β, and MAPK signaling (Figure 3C).

Furthermore, CBD decreased the expression of genes responsible for glucose metabolism, glycolysis (Table 2), and gluconeogenesis, such as *PFKFB4*, which regulates fructose-2,6-bisphosphate and responds to hypoxia to help cancer cells produce more ATP (Figure 3D). There was also downregulation of hexokinase 2 (*HK2*) expression involved in the rapid activation of glycolysis in cancer cells. The transcript for glucose-6-phosphate isomerase (GPI) was decreased, too. The function of GPI, along with the regulation of glucose metabolism, is acting as an autocrine motility factor, a tumor-secreting cytokine, and an inducer of angiogenesis. This shows that the addition of CBD inhibited transcription of factors that help cancer cells in energy and oxygen scarcity and might prevent cancer progression.

Addition of CBD changed the expression of NGF-stimulated transcription kinase and downregulated genes such as *FOSB*, *EGR3*, and *EGR1*. The *FOSB* gene codes for the FosB proto-oncogene, AP-1 transcription factor, which can activate cell proliferation, differentiation, and transformation in cancer cells. *EGR3* and *EGR1*, transcripts for the early growth response 3 and 1, are regulators of cell mitogenesis and differentiation.

CBD showed modulation of TGF-β signaling pathways, including genes such as *SMAD7* and *SMAD3*. SMAD7 was shown to antagonize signaling by the TGF-β type 1 receptor and inhibit TGF-β by associating with their receptors and preventing SMAD2 access. The upregulation of SMAD3, which is part of the transcription factor complex and can serve as a tumor suppressor, has a similar effect. There was also downregulation of gene expression of *IL1RAPL1*, which is a part of the IL-1 receptor family.

Interestingly, CBD decreased the expression of *ABCA1*, the ATP binding subfamily A member 1, which functions as a cholesterol efflux pump and is regulated by the PPAR signaling system. The ABC family transporters are commonly responsible for chemotherapy drug resistance. Thus, CBD might reduce the likelihood of drug resistance development. Furthermore, there was upregulation of *CYP1A1*, a gene coding for monooxygenase enzymes that take part in drug metabolism.

The decreased transcription of factor JUN could inhibit T-cell activation-induced death and reduce transcription in response to Toll-like receptor signaling. JUN transcript is also responsible for KRAS-mediated transcriptional activation of USP28 in CRC. Furthermore, it takes part in the MyD88-dependent cascade initiated by endosome signaling. Additionally, CBD increased the expression of CD22, which helps in B-cell activation. Finally, there was a decreased level of *MAP3K8* transcript, an oncogene coding for mitogen-activated protein kinase 8, which can activate both MAP and JNK pathways as well as pro-inflammatory pathways involving activation of NFκB.

Based on Reactome data for differential expression analysis in HCT-116 CRC cells, CBD had various effects on gene expression. It decreased the expression of genes responsible for glucose metabolism, glycolysis, and gluconeogenesis, which may reduce tumor growth. It could also downregulate cell mitogenesis and differentiation genes. Additionally, CBD modulated pathways involving TGF-β signaling. This could decrease the likelihood of the development of drug resistance. In summary, these findings suggest that CBD may have potential therapeutic benefits for treating CRC. However, more experimental data would be needed to support this notion.

### 2.4. Intermittent Serum Starvation Changed the Transcription of Genes Responsible for Lipid Metabolism and Cell Survival Pathways in the HCT-116 CRC Cell Line

Based on dot plot analysis, ISS changed multiple pathways involved in CRC cell survival, lipid metabolism, extracellular matrix organization, ERBB2 receptor signaling, as well as the PI3K/AKT pathway (Figure 3E).

There was a strong upregulation of multiple genes that participate in NTRK1 signaling, including *ARC*, *EGR1*, *EGR2*, *EGR3*, *EGR4*, *FOS*, *FOSB*, *FOSL1*, *ID2*, *ID3*, *ID4*, *JUNB*, *JUND*, *NAB2*, *SRF*, and *TRIB1* (Figure 3F). Most listed factors participate in cell proliferation, invasion, metastasis, and resistance to chemotherapy-induced apoptosis. The increased levels of *EGR1–4*, the early growth response genes involved in regulating cellular proliferation, differentiation, and survival, and the increased level of serum response factor (SRF), which is a transcription factor that regulates gene expression in response to extracellular signals, could indicate the activation of pro-survival mechanisms in tested cancer cells.

On the other hand, there was a downregulation of *TF* (coding for iron-binding factor) and *VGF*, a neuropeptide precursor implicated in cancer progression and metastasis. Downregulation of TF and VGF can have different effects on CRC, with TF downregulation potentially leading to reduced proliferation and increased apoptosis, while VGF downregulation may decrease migration and invasion of CRC cells.

There were multiple genes that were upregulated under the RAF-independent MAPK1/3 activation term (Table 2), which involved *DUSP1*, *DUSP10*, *DUSP2*, *DUSP4*, *DUSP5*, *DUSP6*, and *DUSP8*. The upregulation of the listed *DUSP* genes in CRC can have different effects on MAPK signaling and may contribute to the development and progression of the disease. For instance, the upregulation of DUSP6 may inhibit ERK1/2 activity and prevent apoptosis. DUSP8 is a phosphatase that dephosphorylates and deactivates JNK and p38 MAPKs, and it was shown to inhibit cell proliferation and migration of cancer cells. However, the upregulation of DUSP2 has been shown to have both pro- and anti-tumorigenic effects in different cancer types. Overall, the effects of upregulating DUSP in CRC can be complex and may depend on the specific DUSP isoform.

Multiple genes responsible for interferon signaling, including *EGR1*, *GBP2*, *GBP5*, *IFI35*, *IFI6*, *IFIT1*, *IFIT3*, *IFIT5*, *MX1*, *OAS1*, *OAS3*, *OASL*, *SMAD7*, *STAT1*, and *UBE2L6,* were also upregulated. Interferon signaling is thought to have anti-tumor effects by promoting apoptosis and inhibiting tumor cell proliferation and migration. On the other hand, some upregulated genes, such as *STAT1* and *SMAD7*, could have pro-tumor effects. STAT1 is a transcription factor involved in immune system activation in response to viral infection, and it has been shown to promote tumor growth and survival. Additionally, SMAD7 is involved in the TGF-β signaling pathway and can promote tumor growth and metastasis.

Another interesting effect of ISS was changes in the regulation of lipid metabolism via PPARα signaling. The expression of genes responsible for making medium-chain acyl-CoA dehydrogenase (ACADM) and supporting long-chain fatty acid synthesis and cholesterol metabolism (including the *HMGCS1* gene, which is also involved in the synthesis of ketone bodies) was increased. There were a few genes that were downregulated, too. Decreased levels of *ANGPTL4*—coding for angiopoietin-like protein 4, which regulates lipid and glucose metabolism and the growth of new blood vessels—could lead to inhibition of angiogenesis in cancer cells. Interestingly, there was a decreased expression of *NR1D1* (nuclear receptor subfamily 1 group D member 1), which is involved in the regulation of circadian rhythms by repressing the expression of core clock components. *NR1D1* also regulates lipid metabolism, adipogenesis, gluconeogenesis, and the macrophage inflammatory response.

The genes *BTC*, *EREG*, *HBEGF*, and *NRG2,* which take part in the epidermal growth factor receptor family, were upregulated under ISS. On the other hand, the decreased expression of *EGFR* (epidermal growth factor receptor), which takes part in the regulation of cell growth, proliferation, differentiation, and survival, could be one of the important targets in cancer therapy. Additionally, decreased levels of *NRG1* transcript could decrease proliferation and survival. Moreover, lowering *PRKCA* (protein kinase C alpha) and *PTK6* (protein tyrosine kinase 6) could also inhibit cell growth, proliferation, and cancer progression.

The following genes were overexpressed in PI3K signaling under ISS treatment: *AREG* (amphiregulin), *BTC* (betaleucin), *EREG* (epiregulin), *FGF9* (fibroblast growth factor 9), *HBEGF* (heparin-binding EGF-like growth factor), *NRG2* (neuregulin 2), *PIK3AP1* (phosphoinositide-3-kinase adaptor protein 1), and *PIK3R3* (phosphoinositide-3-kinase regulatory subunit 3). All the mentioned genes take part in the activation of cell growth, proliferation, insulin signaling, and cancer progression, which could result in unwanted effects of ISS in cancer. On the other hand, *EGFR*, *NRG1*, and *PDGFA* were downregulated under ISS. The epidermal growth factor receptor, coded by the *EGFR* gene, triggers a signaling pathway that promotes cell growth and proliferation of cancer cells. Thus, inhibition of EGFR by ISS could be beneficial in CRC therapy.

Under ISS treatment, many genes that code for the extracellular matrix organization were upregulated, including *BMP4*, *CAPN8*, *COL3A1*, *COL5A2*, *FBLN5*, *ITGA1*, *SERPINE1*, *SPOCK3*, *THBS1*, and *TLL1*. The bone morphogenic protein encoded by the *BMP4* gene, which plays a role in cell differentiation and proliferation, was overexpressed under ISS. *CAPN8*, coding for calpain 8 protein, a calcium-dependent cysteine protease, plays a role in cytoskeletal organization. The increased expression of genes encoding collagen type 3 and type 5 could indicate stimulation of connective tissue formation by cancer cells. Increased expression of the *FBLN5* gene encoding fibulin 5 could help promote the adhesion of endothelial cells and help develop new arteries. Moreover, *ITGA1* codes for an alpha-1 subunit of integrin receptors. *SERPINE1* (plasminogen activator inhibitor 1) is responsible for inhibiting fibrinolysis. Thus, an increased expression of the listed genes could help with the reorganization of extracellular tissue to the cancer cells’ advantage.

At the same time, multiple gene transcripts taking part in extracellular matrix organization were decreased. These included ADAMTS16, COL13A1, COL4A3, CTSK, DDR2, EFEMP2, FBLN2, FBN1, ITGA10, ITGA5, LAMA4, LAMB3, LAMC1, LAMC2, LOXL2, LTBP3, MMP16, P4HA1, P4HA2, PDGFA, PLOD1, and PRKCA. The ADAMTS16 gene encodes ADAM metallopeptidase with thrombospondin type 1. COL13A1 and COL4A3 encode collagen types 13 and 4, respectively. Collagen type 4 is a main structural component of basement membranes and helps with the formation of new vessels. CTSK encoding cathepsin K, a lysosomal cysteine proteinase that could contribute to tumor invasiveness, was also decreased. DDR2 is a gene encoding the discoidin domain receptor subclass of the receptor tyrosine kinase protein family. This protein is a collagen-induced receptor that activates signal transduction pathways involved in cell adhesion, proliferation, and extracellular matrix remodeling. EFEMP2, a gene encoding EGF-containing fibulin extracellular matrix protein 2, is part of many ECM proteins involved in elastic fiber formation and connective tissue development; specifically, it participates in terminal differentiation and maturation of smooth muscle cells. FBLN2 codes for fibulin 2, an ECM protein that plays a role in organ development and can bind to fibrillin. The decreased levels of FBN1, encoding fibrillin 1, an ECM glycoprotein that is a structural component of calcium-binding microfibrils, could lead to the inhibition of cancer cell invasion. ITGA10 and ITGA5 code for integrin subunit alpha 10 and 5, which are collagen-binding proteins involved in cell-matrix adhesion. LAMA4 (laminin-8 subunit alpha), LAMB3 (laminin B1k chain), LAMC1 (laminin subunit gamma 1), and LAMC2 (laminin subunit gamma 2) were decreased, too. Laminins, a family of ECM glycoproteins, are one of the main noncollagenous constituents of basement membranes, which are implicated in cell adhesion, differentiation, migration, and metastasis. Decreased LOXL2, encoding lysyl oxidase-like 2 delta E13 (essential to connective tissue biogenesis, which catalyzes the first step in forming crosslinks in collagen and elastin), could contribute to the inhibition of cancer invasiveness. LTBP3 (latent TGF-β binding protein 3), which encodes a protein that forms a complex with TGF-β and plays a structural role in the extracellular matrix, was also reduced. MMP16 (matrix metalloproteinase 16) takes part in the breakdown of ECM, tissue remodeling, and metastasis. P4HA1 and 2 (prolyl 4-hydroxylase, alpha polypeptide 1 and 2) encode a key component of prolyl hydroxylase, a major enzyme in collagen synthesis. PDGFA (platelet-derived growth factor subunit A) encodes the protein that is a part of PDGF, which is essential for cell proliferation, cell migration, survival, angiogenesis, and chemotaxis. PLOD1 (procollagen-lysine 1,2-oxoglutarate-dioxygenase) catalyzes the hydroxylation of lysyl residues in collagen-like peptides. PRKCA (protein kinase C alpha) belongs to a family of serine–threonine protein kinases that can be activated by calcium. Its protein product takes part in cell adhesion, cell transformation, cell cycle checkpoint, cell volume control, tumorigenesis, and angiogenesis.

Overall, the reduction of the vast majority of factors that take part in CRC’s invasion and metastatic mechanisms could represent a beneficial effect of ISS. Additionally, activation of lipid metabolism could be beneficial when ISS is combined with CBD, which has been shown to strongly suppress the metabolism of carbohydrates.

### 2.5. The Antagonistic Effect of CBD and Cisplatin Could Be Attributed to the Activation of the Cell’s Pro-Survival Mechanisms in the HCT-116 Cell Line

Based on cell viability data, we established that combining cisplatin with CBD had an antagonistic interaction (Figure 2A). Differential gene expression analysis of cisplatin vs. vehicle control and cisplatin combined with CBD vs. vehicle control revealed that in cisplatin alone, the fold change of transcripts responsible for p53-mediated apoptosis (Figure 3A) was much higher compared to combined cisplatin and CBD (Figure 3G,H) in the HCT-116 cell line. This could be due to CBD and cisplatin interacting in an antagonistic manner in the HCT-116 cell line.

In addition to the results for p53 regulation of apoptosis, the results based on the Reactome dot plot showed the differential gene expression in the TGF-β signaling, extracellular matrix organization, circadian clock regulation, and checkpoints defects (Table 3).

The *BHLHE40* (basic helix–loop–helix family member E40) gene that controls the body’s circadian clock was downregulated. Also, the expression of *NFIL3* (nuclear factor, IL-3 regulated), a transcriptional regulator of genes responsible for the circadian clock, was increased. This might indicate changes in the regulation of circadian rhythms under the combination of CBD and cisplatin.

Moreover, there was an increased expression of genes responsible for the extracellular matrix organization and cancer cell invasiveness, including collagen, laminins, and FGF2 activation, in response to CBD and cisplatin. On the other hand, the increased expression of SMAD7 could prevent the activation of TGF-β and inhibit EMT in cancer cells.

Cisplatin is known to activate apoptosis due to its damaging effect on the DNA. However, our results showed that the increased levels of cyclin E2 (*CCNE2*) and *CDC25A* (cell division cycle 25A) and decreased transcripts for CDK inhibitor 1C (*CDKN1C*) could stimulate cell cycle progression even with damaged DNA. This could also be one of the reasons why CBD with cisplatin had antagonistic interaction.

Despite the upregulation of many genes responsible for cell survival, we observed increased levels of pro-apoptotic transcripts, including *BAX*, *FAS*, *TNFRSF10C*, *TP53I3*, *TP53INP1*, and *TRIAP*. These genes were upregulated in a similar way under cisplatin treatment alone. However, as we already mentioned, when we looked at the dot plot analysis, we observed lower fold enrichment for the CBD and cisplatin combination compared to cisplatin alone. Furthermore, the decreased levels of FOS and JUN, the AP-1 transcription factor subunits, indicate inhibition of cell proliferation.

Based on the Reactome data for differential expression analysis in HCT-116 CRC cells, CBD in combination with cisplatin decreased the expression of genes responsible for glucose metabolism, glycolysis, and gluconeogenesis, similar to CBD treatment alone. Increased expression of *PFKFB4* regulates fructose-2,6-bisphosphate and responds to hypoxia to help cancer cells produce more ATP. There was also a reduction in hexokinase 2 (*HK2*) expression, which is involved in the rapid activation of glycolysis in cancer cells. The transcript for glucose-6-phosphate isomerase (*GPI*) was decreased, too. Along with the regulation of glucose metabolism, the function of GPI is to act as an autocrine motility factor, a tumor-secreting cytokine, and an inducer of angiogenesis. This shows that the addition of CBD inhibited transcription of factors that help in the survival of cancer cells under energy and oxygen scarcity and might prevent cancer progression.

Overall, based on our mRNA expression results, we suggest that the antagonistic effect of combining CBD and cisplatin was the result of lower fold enrichment in genes responsible for apoptosis when compared to cisplatin alone in the HCT-116 CRC cell line. However, our data would need to be confirmed with additional experiments involving other cell viability and apoptosis assays, as well as protein expression analysis for selected genes.

### 2.6. The Moderate Antagonistic Effect of Intermittent Serum Starvation and Cisplatin Could Be Attributed to the Activation of the Cell’s Pro-Survival Mechanisms in the HCT-116 Cell Line

The combination of cisplatin and ISS showed moderate antagonistic effects in the HCT-116 CRC cell line (Figure 2B). However, in other tested cell lines, this combination was synergistic. When we looked at mRNA expression data (Figure 3I,J) in HCT-116, the addition of ISS to cisplatin decreased the expression of p53-mediated pro-apoptotic signaling, which was one of cisplatin’s effects toward pro-survival mechanisms. We speculate that because ISS could reduce the speed of DNA replication by slowing the cell metabolic rates, the important stage of the cell cycle for cisplatin’s cytotoxic effects, it could have had reduced anticancer effects under combinational treatment.

Furthermore, the elevated levels of brain-derived neurotrophic factor (*BDNF*), early growth response proteins 1-4 (*EGR1-4*), and parts of AP-1 transcription factors (*FOS* and *JUNB*) indicated activation of cell survival mechanisms and cell proliferation. Increased levels of BCL2 indicate inhibition of apoptosis. A higher SRF could also stimulate cell proliferation. *SRF* is a part of the immediate-early genes that regulate the cell cycle, and apoptosis that couples cellular gene expression to cytoskeletal dynamics; upregulation of *SRF* could indicate strong pro-survival signals in cancer cells. Moreover, increased levels of *TP53I3* (encoding tumor protein P53 inducible protein 3) and its effector *GADD45A* indicated possible activation of p53-mediated DNA repair.

The elevated *HBEGF* and *PDGF* indicated stimulation of growth factor signaling. Additionally, higher levels of *MAP2K3* could indicate the activation of MAPK signaling, which is necessary for glucose transporter expression. Interestingly, lower levels of pyruvate dehydrogenase (encoded by *PDK1)*, a mitochondrial enzyme that catalyzes the oxidative decarboxylation of pyruvate and regulates the homeostasis of carbohydrate fuels in mammals, would suggest inhibition of the Krebs cycle and oxidative phosphorylation in HCT-116 cells.

The increased levels of *BMP4* (bone morphogenic protein), *CEACAM1* (CEA cell adhesion molecule), COL5A2 (collagen type V alpha 2 chain), *LAMA2* (laminin subunit alpha 2), *NTN4* (netrin 4), *PDGFB* (platelet-derived GF subunit B), SERPINE1 (serpin family E member 1), *SPOCK3* (osteonectin), and *THBS1* (thrombospondin 1) indicated a wide variety of responses related to interactions with the extracellular matrix, which can represent an attempt of cancer cells to activate its invasive potential. Furthermore, the elevated levels of *ARC*, a gene coding for an activity-regulated cytoskeleton-associated protein involved in cell migration, could lead to cancer cell progression. Another protein regulating microtubular and vesicular transport that could cause cancer cell progression was dynamin 3 (*DNM3*); the expression of *DNM3* was also increased.

However, with the lowering the expression of genes coding for proteins that take part in Ras signaling, including CRK-like proto-oncogene adaptor protein (*CRKL*) and Ras-like without CAAX 1 (*RIT1*), the GTPase regulating the p38 MAPK-dependent signaling cascade that responds to cellular stress, as well as *MAPK7*, could lead to the reduction of cancer cell proliferation and inhibition of its survival. Moreover, decreased levels of transcripts responsible for the formation of some collagens, laminins, and metalloproteases could lead to the suppression of cancer cell invasiveness.

The increased transcripts for *EGR1*, interferon-induced proteins (*IFI6*, *IFIT1*, *IFIT3*, *IFIT5*), and MHC class II (HLA-DRA) indicated strong stimulation of IFN signaling, which could result in activation of macrophages in the tumor microenvironment. Additionally, the listed transcripts participate in the negative regulation of apoptosis and inhibition of cell proliferation by activating p21 and p27.

Based on the mRNA expression results, we observed an activation of multiple pro-survival and pro-invasive mechanisms under the combination of ISS and cisplatin in HCT-116. Thus, we would not recommend this treatment combination in MMR-deficient CRC. However, we would suggest conducting further experiments to support this notion.

### 2.7. The Strong Synergism between Intermittent Serum Starvation and Cannabidiol Could Be Attributed to the Suppression of Carbohydrate Metabolism in the HCT-116 Cell Line

Based on the Reactome dot plot, the genes that take part in MAPK and PI3K/AKT signaling, carbohydrate metabolism, NGF signaling, and apoptosis were differentially expressed when we compared the combination of CBD and ISS to the DMSO control in the HCT-116 CRC cell line (Figure 3K).

When we analyzed each major Reactome term for differentially expressed genes, multiple pathways of interest were changed (Figure 3L). The following genes under the “TP53 regulate the transcription of cell death genes” term increased in their expression—*FAS*, *PERP*, *PMAIP1*, and *TP53I3*. All the mentioned genes might help with stimulating apoptosis. For instance, *FAS* codes for the cell surface death receptor, a TNF superfamily of receptors taking part in the extrinsic apoptotic pathway. *PERP* is a p53 apoptosis effector related to PMP22. Also, *PMAIP1* (phorbol-12-myristate-13-acetate-induced protein 1) is a pro-apoptotic subfamily within the BCL-2 protein family that determines whether a cell commits to apoptosis. Lastly, *TP53I3* is involved in p53-mediated cell death and could further assist in the suppression of CRC cell growth.

Similar to CBD alone, the combination of CBD and ISS inhibited multiple genes responsible for carbohydrate metabolism. The inhibition of hexokinases 1 and 2 and other enzymes that are part of glycolysis and gluconeogenesis was downregulated. It was shown that many tumors rely on glycolysis as a major source of energy production. Thus, inhibition of carbohydrate metabolism could drastically affect cancer cells’ survival [74].

There was also an increase in the expression of RAF/MAP kinase cascade genes, including *AREG*, *BTC*, *DUSP1*, *DUSP2*, *DUSP4*, *DUSP5*, *EGF*, *FGF19*, *FGF2*, *FGF9*, *HBEGF*, *ICMT*, *PDGFB*, *PSMB10*, and *PSMB9*. The increased levels of amphiregulin (*AREG*), epidermal growth factors, fibroblast growth factors 19, 2, and 9, and platelet-derived growth factor subunit B transcripts, which play essential roles in the stimulation of cancer cell proliferation, survival, EMT, and invasion, could be responsible for cell survival under CBD and ISS treatments. Increased transcripts of heparin-binding EGF-like growth factor (*HBEGF*), a positive regulator of the epidermal growth factor receptor, and protein kinase B gene might overstimulate stress response pathways and cell survival in a way that could be damaging to cancer cells. Additionally, there was an increased expression of dual-specificity phosphatases 1, 2, 4, and 5 (*DUSP1*, *DUSP2*, *DUSP4*, *DUSP5*), negative regulators of MAPK1/ERK2 SAPK/JUN, p38, which act result in response to environmental stress and negative regulation of cellular proliferation. Also, the downregulation of fibroblast growth factor receptors 3 and 4 (*FGFR3* and *FGFR4*, which are genes coding for cell surface receptors that respond to fibroblast growth factors and regulate cell proliferation, migration, lipid metabolism, glucose uptake, and activation of RAS, MAPK1/ERK2, and MAPK3/ERK1, as well as AKT1 signaling cascades) would result in inhibition of cell proliferation. Furthermore, the decreased expression of kinase suppressor Ras 2 (*KSR2*) and RAS guanyl-releasing protein 1 (RASGRP1), which promotes BRAF-mediated phosphorylation of MAP2K1/MEK1 and helps in the activation of Ras, would also contribute to the inhibition of cell proliferation. Additionally, inhibition of fibronectin 1 (*FN1*), involved in cell adhesion, migration, and invasion of cancer cells, could suppress tumor progression.

Next, we analyzed changes in the “PI3K/AKT signaling in cancer” term. The increased levels of transcripts for a component of inhibitor of NFκB (*CHUK*) and NFκB inhibitor alpha (*NFKBIA*) could suggest inhibition of inflammatory cytokine formation. Phosphoinositide-3-kinase regulatory subunit 3 (*PIK3R3*), a regulatory subunit of PI3K, was also increased, which could point to the overactivation of the PI3K/AKT pathway. Moreover, the higher levels of the nuclear receptor subfamily 4 group A member 1 (*NR4A1*) gene transcript, encoding the steroid–thyroid hormone–retinoid receptor protein, a transcription factor induced by the serum stimulation, could further indicate an attempt of cancer cells to stimulate cell proliferation. On the contrary, decreased levels of neuregulin 1 (*NRG1*), a direct ligand of ERBB3 and ERBB4 tyrosine kinase receptors, which activate ERBB receptors, could be responsible for the inhibition of cancer cell growth. Additionally, the downregulated fibroblast growth factor receptors 3 and 4, PDGFA, and GAB1 (GRB2-associated binding protein 1), which take part in FGF signaling, indicate downregulation of this pathway and possible suppression of tumor cell invasiveness and EMT.

The increased signaling by the “NTRK1 (neurotrophic tyrosine receptor kinase gene fusion)” term could promote cancer cell growth. The elevated levels of *BDNF* (brain-derived neurotrophic factor), *EGR 1-4* (early growth response proteins 1-4), *JUNB* (transcription factor jun-B), which enables sequence-specific dsDNA binding activity and is a part of AP-1 transcription factors, and *SRF*, which stimulates cell proliferation and takes part in immediate-early genes, might indicate stimulation of cell proliferation. Moreover, increased *TIAM* (T-lymphoma invasion and metastasis-inducing protein 1), a guanine nucleotide exchange factor (*GEF*) that regulates RAC1 signaling pathways affecting cell shape, migration, adhesion, growth, survival, actin cytoskeleton formation, endocytosis, and membrane trafficking, could contribute to activation of cancer cell migration and metastasis.

On the other hand, lower levels of ARC (activity-regulated cytoskeleton-associated protein) involved in cell migration and cytoskeleton organization may indicate decreased cancer cell invasion. Additionally, decreased *CRKL* (CRK-like proto-oncogene adaptor protein), which has been shown to activate the RAS and JUN kinase signaling pathways and transform fibroblasts in a RAS-dependent fashion; *MAPK7* (mitogen protein kinase 7), involved in cell proliferation and transcription regulation; and *RIT1* (Ras-like without CAAX 1—Ras-related GTPase), involved in regulating p38 MAPK-dependent signaling cascades related to cellular stress, could support the hypothesis that a combination of CBD and ISS can have an anti-proliferative effect.

Based on the “IL-1 signaling” term, there was an overall inhibition of NFkB signaling. As we already mentioned, increased expression of an NFkB kinase complex (*CHUK*) inhibitor component and *NFKBIA* suggests the downregulation of NFkB signaling. Higher levels of NLR family CARD domain containing 5 (a caspase-recruitment-domain-containing NLR family playing an important role in the cytokine response through inhibition of NFkB) and negative regulation of IFN signaling could also contribute to inhibiting pro-inflammatory cascades. Additionally, the downregulation of *IL1RAP* (IL1 receptor accessory protein), which helps in the activation of signaling events of IL-1-responsive genes *NFKB2*, *NOD1*, *NOD2*, members of Nod1/Apaf-1 family proteins that take part in the immune response to LPS and activation of NFkB, and TRAF6 (TNF receptor-associated factor 6, which helps in the activation of NFkB and response to inflammation) suggests that NFkB signaling was substantially downregulated under our treatment.

In CRCs, the RAS-MAPK pathway is overactivated, with KRAS and BRAF being overexpressed in approximately 50% and 15% of cases [37,38]. PI3K/AKT signaling is upregulated in almost 40% of colon malignancies [39]. When AKT is activated, it can increase enzymes in the glycolytic pathway to generate ATP, and some of the byproducts of glycolysis are utilized for cancer cell growth.

Based on our results, our hypothesis behind the synergistic interaction between ISS and CBD is that ISS overactivated PI3K/AKT signaling, which, in part, made cancer cells rely on glycolysis; it should be noted, however, that CBD strongly inhibited carbohydrate metabolism. As a result, it is possible that cancer cells were not able to generate enough ATP via pro-survival pathways, resulting in the stimulation of cell death transcripts.

### 2.8. The Combination of Intermittent Serum Starvation, Cisplatin, and Cannabidiol Resulted in the Strong Upregulation of Transcripts Responsible for G2/M Cell Cycle Arrest in the HCT-116 Cell Line

Under the combination of ISS, cisplatin, and CBD, there was a massive decrease in transcripts responsible for the resolution of sister chromatid cohesion and in genes involved in metaphase and anaphase, sister chromatids, and amplification of signals from kinetochores (Figure 3N, Table 3).

The dot plot analysis showed a strong upregulation of genes responsible for sister chromatid cohesion, occurrence of mitotic spindle checkpoint, amplification of signals from kinetochores, and activation of TP53-mediated apoptosis and disorders of carbohydrate metabolism (Figure 3M). We suggest that the combination of CBD and ISS caused strong inhibition of energetic reserves in CRC cells, which could assist in cisplatin’s cytotoxic effect. We already mentioned that cisplatin often causes cell cycle arrest in the G2/M phase [48], which we also observed in pathway analysis.

The increased levels of *CDKN1A*, a gene transcript coding for cyclin-dependent kinase inhibitor 1A, could indicate activation of cell cycle arrest. The decreased levels of *AURKB*, aurora B kinases responsible for the alignment and chromosome segregation during mitosis through association with microtubules, could also indicate cell cycle arrest in the M phase. Moreover, decreased *BUB1*, a gene coding for mitotic checkpoint serine/threonine kinase B that has been localized in kinetochores and plays an important role in inhibiting the anaphase-promoting complex/cyclosome (APC/C), could cause delays in the onset of anaphase. The decreased levels of centromere proteins A, F, E, I, and L would disturb the regular mitotic behavior of chromosomes, leading to improper centromere structures. A decrease in *CLASP1*, a nonmotor microtubule-associated protein involved in regulating microtubule dynamics at the kinetochores, also indicated a disturbance of proper mitosis. Lower levels of *NUF2*, the component of the NDC80 kinetochore complex, would suggest improper chromosome segregation and spindle checkpoint activity. Additionally, *MAD2L1* (mitotic arrest deficient 2 like 1, a component of the mitotic spindle assembly checkpoint that prevents anaphase onset until all chromosomes are properly aligned) was decreased, suggesting arrest in the M phase. *PLK1* (coding for polo-like kinase 1, highly expressed during mitosis and regulates centrosome maturation and spindle assembly, removes cohesins from chromosome arms, inactivates APC/C inhibitors, and regulates mitotic exit and cytokinesis) was also decreased. Furthermore, decreased expression of *RAD21*, a gene involved in the repair of DNA double-strand breaks and chromatin cohesion during mitosis, would suggest disturbance of the homologous recombination repair system.

Moreover, the lowering of cyclin-dependent kinase 1 (*CDK1*), cell division cycle 20 (*CDC20*), and cyclin B1 and B2 (*CCNB1* and *CCNB2*) would suggest strong inhibition of cell cycle progression in the G2/M phase.

Similar to cisplatin alone, p53-mediated pro-apoptotic factors such as FAS and BAX transcripts were elevated. Additionally, decreased levels of *survivin* and *NDRG1* (encoding N-myc downstream regulated 1, a stress response protein involved in hormone response) would further suggest inhibition of pro-survival mechanisms and activation of apoptosis.

At the same time, multiple transcripts responsible for the tumor’s cell survival mechanisms were also kicking in. There were increased levels of *JUNB* protooncogene, an AP-1 transcription factor subunit that regulates cell proliferation. Additionally, the increased levels of *HIF1α* and *FGF2* indicated activation of survival pathways in cancer cells and possible stimulation of EMT, which would constitute unwanted effects of combinational treatment. Additionally, increased Ras homolog family member U, a Rho GTPase activating PAK1 and JNK1, can induce the formation of filopodium and dissolving stress fibers. It also mediates the effects of Wnt signaling in regulating cell morphology, cytoskeletal organization, and cell proliferation. Moreover, there was an increase in *TNFRSF1B*, a gene encoding a member of the TNF-receptor superfamily. Together with *TNFR1*, this forms a heterocomplex that mediates the recruitment of c-IAP1 and c-IAP2, which could help in cancer cell survival.

The increased betacellulin (*BTC*) levels could indicate the increased production of EGF-like proteins to stimulate cancer cell growth. Additionally, higher levels of *FGF1*, *FGF2*, *FGF9,* and *FGF19* indicate a strong stimulation of FGF signaling, which could stimulate broad mitogenic and cell survival mechanisms, including angiogenesis, invasion, and metastasis. When elevated, heparin-binding EGF-like growth factor (HBEGF) could stimulate PI3K/AKT pro-survival signaling. Moreover, elevated levels of PDGF, TGF-α, and PI3K regulatory subunit 3 also indicated activation of PI3K/AKT signaling.

On the other hand, reduced expression of fibroblast growth factor receptor 4 (*FGFR4*) and *EGFR* could decrease cell proliferation, migration, and lipid and glucose metabolism. Lower levels of IL-1 receptor accessory protein could reduce IL-1-dependent activation of NFkB and result in an anti-inflammatory effect. Additionally, *SNAIL1*, a strong inducer of EMT, was also decreased, which would indicate inhibition of tumor cell progression. Decreased expression of *RPAGD* (encoding Ras-related GTP binding D, which takes part in AKT/mTOR signaling and activation and relocation of mTORC1 to the lysosomes) could inhibit CRC cell growth.

The increased expression of different types of collagen components such as collagen type I alpha 1 chain, collagen type IV alpha 5 chain, and collagen type XI alpha 2 chain, as well as CEA adhesion molecule 1, integrin subunits alpha 1 and 10, laminin subunits, and syndecan 1, indicated an attempt of tumor cells towards the formation of ECM connections between cancer cells and the environment, which could lead to invasion and metastasis. However, the downregulation of *ADAMTS2*, *ADAMTS14*, *MMP11*, and *MMP7*, metalloproteases that help in cancer cell invasiveness by dissolving the ECM components, could prevent cancer invasive potential.

Overall, we observed a strong activation of G2/M cell cycle arrest and pro-apoptotic transcripts in HCT-116 CRC cells under the combination of CBD, cisplatin, and ISS. The possible downside could be that despite the strong cytotoxic effects of combinational treatment, there was an upregulation of transcripts that take part in cancer cell progression and activation of cancer’s invasive potential.

## 3. Materials and Methods

### 3.1. Main Reagents

Cisplatin was obtained from Sigma-Aldrich (Oakville, ON, Canada, CAS 15663-27-1), and CBD was also sourced from Sigma-Aldrich (Cerilliant, C-045 Lot: FE01271601). DMSO (Dimethyl sulfoxide anhydrous) was purchased from Thermo Fisher Scientific (High River, AB, Canada, Cat#D12345). CBD (10 mg/mL) was dissolved in methanol and stored at −20℃. The stock solution of cisplatin (100 μM) was dissolved in DMSO and kept at −20℃ for no longer than three weeks.

### 3.2. Cell Culture and Maintenance

The experiments were performed on three human CRC cell lines, HT-29 (HTB-38™), HCT-116 (CCL-247™), and LS-174T (CL188™), and immortalized human colonic epithelial cell line HCEC-1CT (abm catalogue N# T0715). The CRC cell lines HCT-116 (CCL-247™) and LS-174T (CL188™) were purchased from ATCC (Rockvile, MD, USA). HT-29 (HTB-38™) was a kind gift from Dr. Roy Golsteyn’s Laboratory at the University of Lethbridge. All cell lines were cultured according to the manufacturer’s instructions. 

### 3.3. Treatments

#### 3.3.1. Exposure of CRC Cells to Cisplatin

To establish IC50s, a range of concentrations (1–15 μM) of cisplatin was obtained by diluting cisplatin in the fresh complete media, and those were tested on CRC and a normal colon epithelial cell line.

#### 3.3.2. Exposure of CRC Cells to CBD

To establish IC50s, a range of concentrations (2–12 μM) of pure CBD were obtained by diluting the cannabinoids in the fresh complete media, and those were tested on CRC and normal cell lines.

#### 3.3.3. Exposure of CRC Cells to Intermittent Serum Starvation

To maintain the same nutritional supply for each CRC cell line, the cells were cultivated in RPMI-1640 medium (30-2001^TM^). The intermittent serum starvation (ISS) was recreated by depriving cells of FBS for 16 h and reintroducing media with FBS for 8 h. The experiments lasted for five consecutive days in all cell lines. The first combination group included complete media for 24 h with the addition of cisplatin (8 h) and/or CBD (16 h). The second combination group included ISS for 16 h with the introduction of CBD (16 h), cisplatin (8 h), and their combination.

### 3.4. Cell Viability Assay (MTT)

Cell viability was determined via MTT [3-(4, 5-dimethylthiazol-2-yl)-2, 5-diphenyltetrazolium bromide] assay. An MTT assay (Roche, Sigma-Aldrich, Germany) was used to evaluate the effects of cisplatin, CBD, and ISS on CRC and normal epithelial colon cell viability. 

Cells were incubated to 80–90% confluency in 10 cm Petri dishes. Next, cells were trypsinized with trypsin/EDTA (0.25% trypsin and 2.21 mM EDTA-4Na; Cat#325-043-EL; WISENT Inc., Quebec, QC, Canada). After trypsinization and centrifuging, fresh medium was added to the cells, and one (for experiments with complete media) or three (for ISS experiments) thousand cells per well were plated in triplicate into flat-bottomed 96-well plates in 100 μL of the appropriate medium. The cells were allowed to adhere to the plate surface overnight and were exposed to treatment for five days. Cells were then maintained at 37 °C in a humidified atmosphere containing 5% CO_2_ for 24 h. Treatments were changed daily with the normalized concentrations of methanol and DMSO. 

Every 24 h of treatment, 10 μL of MTT kit I (#11465007001, Roche, ON, Canada) was added, and plates were incubated at 37 °C in the CO_2_ incubator for 4 h. After the incubation, 100 μL of MTT solution was added to each well, and the plates continued to incubate at 37 °C overnight. Absorbance was measured at 595 nm using a FLUOstar Omega microplate reader (BMG Labtech Ortenberg, Germany). Results were calculated by comparing the treatments to the appropriate controls. All treatments were in triplicate, and each test was performed in at least three independent experiments. 

### 3.5. RNA Extraction and Gene Expression Analysis

Once grown to 85–90% confluency in a Petri dish, cells were detached using trypsin/EDTA (0.25% trypsin and 2.21 mM EDTA-4Na, Cat#325-043-EL, WISENT INC., Quebec, QC, Canada) and re-plated in 6-well plates at a density of 6 × 10^5^ cells/well. After 24 h of incubation, cells were exposed to treatments for 72 h. The culture medium was replaced with a fresh treatment medium every 24 h and at specific time points (at the 8th and 16th hours during a 24-h period) for ISS experiments in the HCT-116 cell line. At the endpoint, cells were washed twice with ice-cold PBS and harvested using the TRIzol™ Reagent (Invitrogen). Total RNA extraction followed the TRIzol Reagent protocol (https://assets.thermofisher.com/TFS-Assets/LSG/manuals/trizol_reagent.pdf, accessed on 25 April 2023). Immediately after the isolation, the quality and quantity of the extracted RNA were assessed using a NanoDrop 2000/2000c Spectrophotometer (Thermo-Fisher Scientific Company, Wilmington, DE, usa). Additionally, the integrity of RNA samples was assessed using agarose gel electrophoresis. 

The RNA samples from the HCT-116 CRC cell line were shipped to Génome Québec (Montréal, QC, Canada) for mRNA library preparation and next-generation sequencing. The stranded mRNA libraries were prepared using the NEBNext^®^ Ultra™ RNA Library Prep Kit for Illumina^®^ (New England BioLabs). Massive parallel sequencing was performed using the Illumina NovaSeq6000 S4 (PE 100 bp—25 M reads, number of sequencing units—30.00). Base calling and demultiplexing were performed by the sequencing provider (Genome Quebec). Libraries were generated from 250 ng of total RNA using the kit Illumina^®^ Stranded mRNA Prep, Ligation (Illumina), as per the manufacturer’s recommendations. Adapters and PCR primers were purchased from Illumina. Libraries were quantified using the KAPA Library Quantification Kits—Complete kit (Universal) (Kapa Biosystems). The average size fragment was determined using a LabChip GXII (PerkinElmer, Shelton, CT, USA) instrument. The libraries were normalized and pooled and then denatured in 0.05 N NaOH and neutralized using HT1 buffer. The pool was loaded at 175 pM and 200 pM on an Illumina NovaSeq S4 lane using the Xp protocol as per the manufacturer’s recommendations. The run was performed for 2 × 100 cycles (paired-end mode). A phiX library was used as a control and mixed with libraries at the 1% level. Base calling was performed with RTA v3.4.4. The program bcl2fastq2 v2.20 was then used to demultiplex samples and generate fastq reads.

Initial quality control was conducted using FastQC v0.11.9 https://www.bioinformatics.babraham.ac.uk/projects/fastqc/, accessed in May–June 2023. Sequencing reads were trimmed of adapter sequences and low-quality bases using Trimmomatic. Trimmed sequence files were examined with FastQC to verify the trimming results.

Trimmed sequencing reads were mapped to the human genome (GRCh37, Ensembl) downloaded from the Illumina iGenome website (https://support.illumina.com/sequencing/sequencing_software/igenome.html, accessed in May–June 2023). Mapping was performed using splice-aware aligner HISAT2 2.1.0 [75]. Alignment files in SAM format were converted to BAM, sorted and indexed with samtools v.1.3.1 [74]. Mapping quality and statistics were collected with QualiMap software package v.2.2.2 http://qualimap.conesalab.org/, accessed in May–June 2023 [76]. The counts of reads mapping to features (genes) were counted using FeatureCounts v.2.0.1 software [77].

Output summaries and quality reports from various software used in the analysis were integrated into a single report using multiqc v.1.13 https://multiqc.info/, accessed in May–June 2023 [78].

Data exploration, visualization and statistical comparisons were conducted using R language version 4.2.2. Pair-wise comparisons between experimental groups were performed with DESeq2 v.2.1.36 [79], as described in the package manual. To decrease computational time, only the genes with at least 5 reads across 3 samples were kept in the analysis. In addition to the hard threshold filtering mentioned above, DESeq2 implements independent filtering based on the mean of the normalized count as a filter statistic.

We used hierarchical clustering (HC) and principal components analysis (PCA) to investigate the relationship between samples and detect potential outliers. Prior to HC and PCA analysis, DESeq2-normalized values underwent variance stabilizing transformation using the vst() function from DESeq2 [80]. HC was performed using the hclust() function in R, with the clustering method set as “complete” for the matrices of sample-to-sample distances, and “Ward.D2” in case of sample and gene clustering based on the top 500 most variable genes. The distance measure in HC analysis was set to “Euclidean”. Principal components analysis (PCA), applied to the top 500 highly variable genes, was conducted using the prcomp() function in R with the default options.

Differentially expressed genes (DEGs) were detected with the DESeq2 function results() using the default options. DESeq2 uses the Wald test to determine significantly changed genes between groups. The independent filtering option was set to TRUE with the alpha threshold (adjusted *p*-value) kept at 0.1. Multiple-comparison adjustment was performed using the Bejamini–Hochberg procedure. Significantly changed genes were visualized as MA plots, volcano plots and heatmaps built using pheatmap v.1.0.12 and ggplot2 v.3.4.0. The genes involved in the analysis were annotated with Entrez IDs and descriptions using BiomaRt 2.54.0 [81]. Genes with adjusted *p* values below 0.05 and absolute log2 fold change over 0.59 (~1.5 fold change) were considered significant.

Gene ontology (GO) and Reactome pathway enrichment analysis was conducted using pathfindR v.1.64.0, an R package that identifies significantly enriched biological terms utilizing active subnetworks. The main pathfindR function was used with the following options: gene_sets = “GO-All” (use all GO categories), adj_method = “fdr”, enrichment_threshold = 0.05, pin_name_path = “STRING”, search_method = “GR”, max_gset_size = 500. The same options were applied to pathfindR enrichment of Reactome analysis. PathfindR output includes a table with enriched terms, a table of enriched term clusters, dot plots, gene-term maps, and upset plots. Clustering analysis and corresponding plots were prepared using corresponding pathfindR functions with the default settings.

Additionally, pathway topology analysis was performed with the Signaling Pathway Impact Analysis (SPIA) tool (version 2.50.0) [82]. Reactome pathways were used as the reference database. Results with adjusted *p*-values below 0.05 were considered statistically significant.

The analysis of the pathway results, including gene functions and interactions, was performed with the help of GeneCards—the human gene database www.genecards.org, accessed in May–June 2023 [83,84].

### 3.6. Statistical Analysis

GraphPad Prism 9.0 software was used to calculate two- and one-way analyses of variance (ANOVA), and the two-tailed Student’s *t*-test was employed to determine the statistical significance between treatment groups for cell viability experiments. Results were presented as the mean ± SD of data. *p* < 0.05 was considered statistically significant.

### 3.7. Calculation of Combination Index (CI)

To determine synergism, additivity, or antagonism, we performed drug combination analysis based on the Chou–Talalay method using CompuSyn 1.0 software [85,86,87]. In Chou’s approach, the scattering data points fit the median-effect principle with mass-action law. The combination index (CI) was calculated to quantify synergism and antagonism of the drugs, where CI < 1 indicates synergism, CI = 1—additive effect, and CI > 1—antagonistic effect.

## 4. Discussion and Future Perspectives

### 4.1. Cisplatin

Cisplatin is an effective chemotherapy agent that is used for the treatment of multiple cancers [76]. It is often used in combination with other anticancer therapies [45]. Unfortunately, it is possible to develop resistance to cisplatin and, as a result, cancer progression occurs often [46,47].

Cisplatin has a strong pro-apoptotic effect [50,51], which coincided with our HCT-116 CRC cell line data. Cisplatin’s cytotoxic action is explained by the ability to form DNA crosslinks causing DNA damage and apoptosis [88,89]. Additionally, when administered, it induces oxidative stress, primarily in mitochondria, via reduction in membrane potential and ultimately leads to cell death [90].

According to the mRNA expression data obtained from the HCT-116 CRC cell line, cisplatin had multiple effects. It reduced multiple metabolic pathways that cancer cells depend on for growth and proliferation. Cisplatin induced cytotoxic effects by increasing the expression of several genes involved in both the intrinsic and extrinsic pathways of apoptosis. The fold enrichment was one of the highest for TP53 regulation of transcription of cell death genes under cisplatin treatment (Table 2). Our data also showed that multiple unwanted mechanisms were activated under cisplatin’s treatment, which could result in CRC cell survival and progression. These effects are often observed in clinical settings as a disease relapses. Thus, despite the high effectiveness of cisplatin on the HCT-116 CRC cell line, we tested additional treatment combinations to seek synergy and possible reversal of cancer cell progression mechanisms that were kicking in under cisplatin treatment alone. However, to confirm that in parallel to cisplatin’s cytotoxic effects, there was a development of cancer cell survival mechanisms, we would suggest performing single-cell sequencing analysis to reveal the development of subclones resistant to cisplatin’s action.

### 4.2. Cannabidiol

As we previously mentioned, the mechanisms of anticancer effects of cannabinoids include the activation of apoptosis, endoplasmic reticulum (ER) stress response, downregulation of survivin (inhibitor of apoptosis), and a decrease in RAS/MAPK and PI3K/AKT signaling [7,11,16,20,21]. However, our data on the HCT-116 CRC cell line did not show such a variety of mechanisms involved. We suggest that perhaps the dose of IC50 was insufficient to activate the full spectrum of cytotoxic effects of CBD, at least when it was administered alone. It is also possible that the effect of CBD depends on the cancer type and tissues involved.

Adding CBD modulated TGF-β signaling via *SMAD7* and *SMAD3* transcripts’ increase, which resulted in the inhibition of TGF-β, a common pathway (responsible for EMT and cell survival) that is affected in CRCs [91,92]. Interestingly, CBD also inhibited the *ABCA1* transcript, which belongs to the ABC family of transporters that often contribute to chemotherapy drug resistance. Thus, CBD could possibly reduce the development of drug resistance by decreasing drug efflux pumps.

A persistent chronic inflammatory response in some CRCs with increased levels of TNF, IL-17, IL-23, IFN-*γ*, and IL-6 due to the activation of NFkB and STAT3 can lead to the formation of aberrant cryptic foci (ACFs) and adenomas that eventually can develop into adenocarcinomas. Moreover, continuous activation of COX-2 increases KRAS signaling and promotes tumor survival, progression, and metastatic potential [93,94,95]. As our results showed, CBD decreased levels of the *MAP3K8* transcript, an oncogene coding for mitogen-activated protein kinase 8, which can activate both MAP and JNK pathways as well as pro-inflammatory pathways involving activation of NFκB. This could be one of the mechanisms of CBD preventive action on CRCs’ development.

Additionally, we observed the decreased expression of genes responsible for glucose metabolism, glycolysis, and gluconeogenesis, such as *PFKFB4*, which regulates fructose-2,6-bisphosphate and responds to hypoxia to help cancer cells produce more ATP. There was also a decrease in hexokinase 2, which is involved in the rapid activation of glycolysis in cancer cells. Furthermore, our data showed that the addition of CBD inhibited the transcription of factors that help cancer cells with energy and oxygen scarcity. Thus, it could prevent cancer cell survival and the development of cancer progression. The observed mechanisms of CBD regarding glucose metabolism were of high interest to us because they could assist with the mechanisms of serum deprivation when two treatments were combined.

Additionally, CBD downregulated cell mitogenesis and differentiation genes, specifically *FOSB*, *EGR3*, and *EGR1*. CBD had an impact on TGF-β signaling pathways and decreased the expression of ABCA1 and ANGPTL4, potentially reducing the development of drug resistance. Conversely, it increased the expression of oncogenes like *JUN* and *MAPK3K8* transcripts. In summary, these results suggest that CBD may have therapeutic potential for the MSI subtype of CRC. However, further experimental data would be required to support this idea, such as apoptosis assay, protein expression analysis of selected pathways, and animal models.

### 4.3. Intermittent Serum Starvation

Studies have demonstrated that both serum starvation in vitro and short-term food deprivation in vivo can decrease growth factor stimulation levels [58,59,60]. When growth signals are depleted in normal cells, it can reduce proliferation-stimulating signaling activity and metabolism [61]. However, cancer cells respond differently to starvation, whereby it can trigger cellular stress due to their metabolic reprogramming to ensure continuous proliferation [62] and can activate the DNA damage response [63]. In mice models, short-term fasting protected normal cells while sensitizing malignant cells to chemotherapy drugs, which depended on reduced levels of IGF-1 and glucose [96]. This was one of the reasons why we decided to test 0% fetal bovine serum (FBS) for our ISS model. FBS is known to contain a high number of growth-stimulating factors that boost cancer cell growth. We wanted to investigate how pro-survival signaling in CRC cells will respond to ISS because multiple oncogenic pathways play a role in decreased stress resistance in cancer cells, which results in their inability to switch into a stress-protective mode [97].

The mRNA expression analysis of ISSs’ effects in HCT-116 showed that serum deprivation activated multiple survival pathways, including PI3K/AKT signaling. ISS downregulated the expression of EGFR, which could inhibit cell growth, proliferation, and survival of cancer cells. On the other hand, increased levels of serum response factor and PI3K signaling may indicate the activation of survival pathways in CRC cells. Multiple genes that take part in the downregulation of MAPK/ERK1/2 signaling were upregulated, which could lead to the prevention of apoptosis. The numerous genes taking part in extracellular matrix reorganization were changed, too. The decreased levels of some integrins, laminins, MMPs, and PDGFs could point to reduced invasiveness of the HCT-116 cells under ISS treatment. ISS also inhibited carbohydrate metabolism and modulated lipid metabolism similarly to the CBD treatment alone.

Based on the literature review, MTT, and mRNA expression data, we decided to test CBD, cisplatin, and ISS in different combinations to see if we could achieve a synergistic effect. Thus, our next step was to look at different treatment interactions and their molecular mechanisms.

### 4.4. Cisplatin and Cannabidiol

As mentioned previously, the activation of the p38 MAPK pathway plays a crucial role in cisplatin-induced apoptotic death as it triggers the transcription of PUMA and NOXA via p53 activation [50,51]. Our mRNA sequencing results also indicated a strong upregulation of p53-mediated transcription of genes involved in apoptosis, and based on the literature review, these mechanisms were similarly modulated by cannabinoids, which is why we combined cisplatin and CBD to explore the possibility of a synergistic interaction.

Unfortunately, the combination of CBD and cisplatin had an antagonistic effect. Although we observed an activation of pro-apoptotic genes, the fold enrichment of cell death transcripts was much higher in cisplatin alone compared to cisplatin with CBD. We suggest that one of the reasons might be an increased basal activity of pro-survival pathways under CBD treatment due to its modulative role in carbohydrate and lipid metabolism.

### 4.5. Cisplatin and Intermittent Serum Starvation

It was previously shown that serum starvation sensitized cancer cells to cisplatin while protecting normal cells [63]. In normal cells, serum starvation caused cell cycle arrest in the G0/G1 phase due to p53/p21 activation, which depended on AMPK but not on the activation of ATM. However, serum starvation-activated p53 in cancer cells was AMPK- and ATM-dependent. Additionally, the combination of cisplatin with serum starvation led to the activation of the ATM/CHK2/p53 pathway, unlike cisplatin alone, which indicated that the combination therapy sensitized cancer cells to chemotherapy. As a result, short-term starvation sensitized tumor xenografts to cisplatin, as indicated by significant tumor growth delay and the induction of complete remission in 60% of mesotheliomas and 40% of lung carcinoma xenografts. Thus, combining starvation with cisplatin may enhance the therapeutic index of cisplatin-based chemotherapy [63]. However, our results showed different patterns. Based on the cell viability, the combination of ISS and cisplatin had various effects on different CRC cell lines. In the MMR-deficient, p53-positive HCT-116 CRC cell line, we observed antagonistic interaction with the strong activation of pro-survival mechanisms, such as an increase in AP-1 transcription factors, stimulation of growth factor signaling, and upregulation of transcripts that take part in cancer cell invasiveness. On the other hand, in the APC-mutated HT-29 CRC cell line, we observed a synergistic effect. However, without mRNA pathway analysis of the HT-29 cell line, we cannot suggest any molecular mechanisms behind such an effect and would recommend performing further experiments with this treatment combination.

### 4.6. Intermittent Serum Starvation and Cannabidiol

The inhibition of the glycolytic pathway in cancer cells could lead to stress overload due to constant proliferative signaling from oncogenes [71]. Cancer frequently encounters hypoxia-reperfusion in its microenvironment, leading to extreme ROS production that causes mitochondrial damage and apoptosis [72]. Thus, tumors learn to alleviate oxidative damage via increased glycolysis and downregulation of the mitochondrial function [72].

Glycolysis is activated by the PI3K pathway and its downstream target, AKT, which can trigger cell hypertrophy, enhance glycolysis, and activate cell survival [98]. AKT may post-transcriptionally regulate multiple glycolytic steps, such as the localization of glucose transporters in the cell membrane and stimulation of hexokinase function without growth-stimulating signals. High glucose uptake by tumor cells compensates for mitochondrial dysfunction and is required for proliferation. Stimulation of glycolysis enables cells to redirect accumulated pyruvate toward the biosynthesis of lipids, which is needed for membrane assembly [99].

Previous studies showed that many cancer cells have higher glucose uptake rates and rely on glycolysis and lactic acid fermentation even when oxygen is present, known as the Warburg effect [100]. The PI3K/AKT/mTOR pathway is a key regulator of aerobic glycolysis and cellular biosynthesis, involving enhanced glucose uptake, essential amino acids, and protein translation [101]. In cancer, hyperactivation of AKT prevents apoptosis and boosts uncontrolled cell proliferation. The signaling from growth factors and cytokine receptors can inhibit the activation of apoptosis by stimulating PI3K/AKT signaling [102].

The potential of modulating tumor metabolism for therapeutic purposes has not been extensively studied. However, several methods for influencing tumor metabolism could have therapeutic implications. These methods include inhibiting HIF-1α from decreasing angiogenesis, re-establishing p53 to activate apoptosis, and suppressing the PI3K/AKT/mTOR signaling pathway to inhibit cell growth and proliferation [103,104]. Additionally, AKT may inhibit apoptosis via the activation of mitochondria-bound hexokinase [105]. In summary, low-nutrient conditions and glycolysis inhibitors may enhance the activation of apoptosis for glucose-addicted cancer cells with overactivated oncogenes.

Based on our results, we suggest that the strong synergistic effect between ISS and CBD was due to the activation of AKT signaling and simultaneous inhibition of glycolysis and oxidative phosphorylation (Table 3), which pushed cancer cells toward the stimulation of pro-apoptotic factors. However, the major limitation of our study is that we looked only at transcriptomics. Thus, our suggestions need to be supported by other experiments, including those on the protein expression of genes involved in glucose metabolism and the PI3K/AKT pathway.

### 4.7. Intermittent Serum Starvation, Cannabidiol, and Cisplatin

The capacity of cisplatin to create DNA crosslinks results in the activation of cell cycle checkpoints. Cisplatin leads to a temporary pause in the S phase of the cell cycle, aided by p16. Cdc2-cyclin A or B kinase is strongly inhibited, resulting in more noticeable G2/M cell cycle arrest. In addition, cisplatin triggers the activation of ATM and ATR, which includes the phosphorylation of the p53 protein [49]. Our results showed that when CBD, cisplatin, and ISS were combined, we observed synergistic interaction between the treatments. Interestingly, the differential gene expression analysis revealed a strong activation of multiple transcripts taking part in G2/M cell cycle arrest and p53-mediated transcription of cell death genes. We observed a massive decrease in transcripts responsible for the resolution of sister chromatids’ cohesion, genes involved in metaphase and anaphase regulation, and kinetochore signaling. We suggest that ISS and CBD, which acted by suppressing metabolic pathways in the HCT-116 CRC cell line, aided the cytotoxic effects of cisplatin, resulting in the upregulation of pro-apoptotic genes and G2/M arrest. It would be interesting to check if this combination could result in mitotic catastrophe and cell death in multiple CRC cell lines.

## 5. Conclusions and Future Directions

In summary, our work used cell viability outcomes and mRNA expression evaluations to discern treatment synergies within CRC cell lines, shedding light on the underlying molecular mechanisms of these drug interactions. Nevertheless, additional studies are imperative to support our claims. We acknowledge that the drawback of our study is that it was based on the analyses of cell viability and mRNA expression alone. To solidify our hypotheses, additional research is needed. Future studies will have to analyze gene expression at the protein level in specific pathways, conduct apoptosis assays for various treatments, and employ animal models featuring intermittent fasting and drug combinations. Such an approach could yield more dependable data with potential clinical implications.

## Figures and Tables

**Figure 1 ijms-24-14743-f001:**
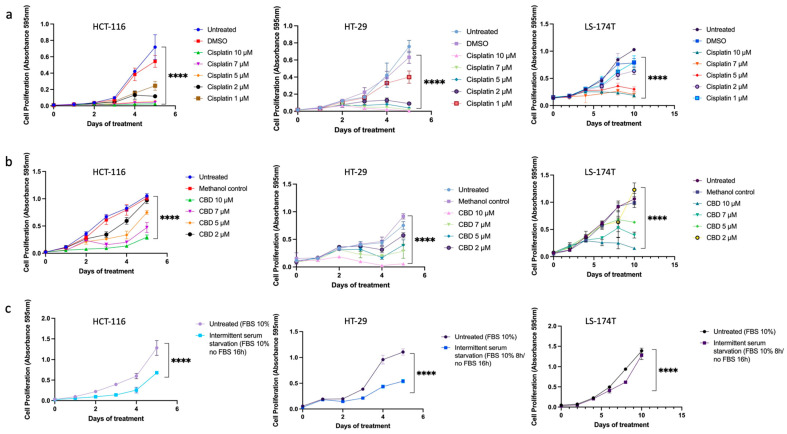
The time and dose-dependent effects of (**a**) cisplatin, (**b**) CBD, and (**c**) ISS on HCT-116, HT-29, and LS-174T CRC cell lines. Results are expressed as means of calculated cell viability ± standard deviations of each group in triplicate at 595 nm absorbance. To calculate time–dose effects, two-way ANOVA was performed using GraphPad Prism version 9.0. Significant differences between groups are marked with **** *p* <0.0001.

**Figure 2 ijms-24-14743-f002:**
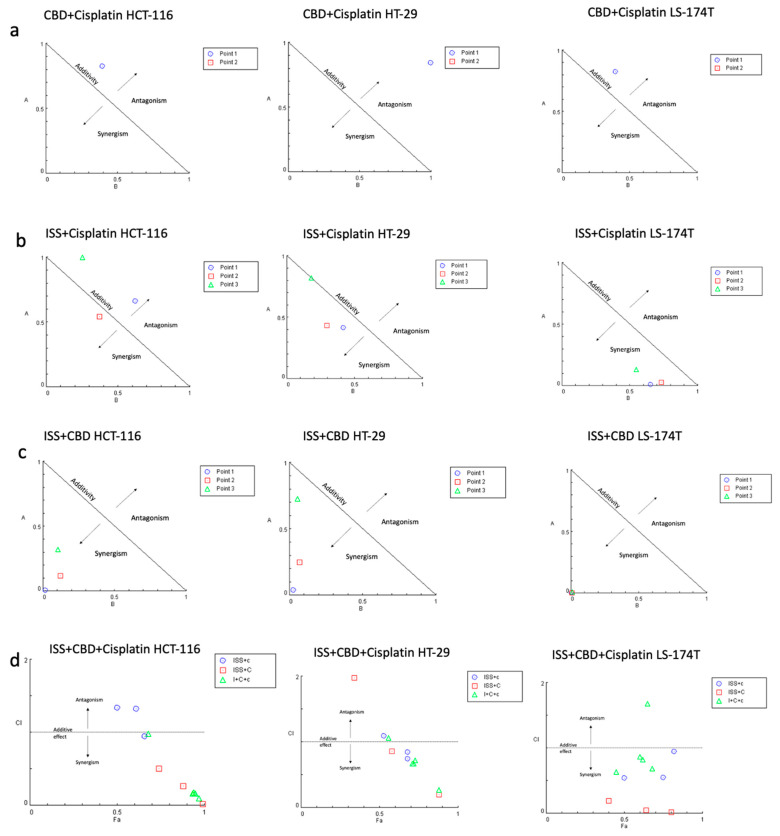
The combination effects of (**a**) CBD and cisplatin; (**b**) cisplatin and ISS; (**c**) CBD and ISS; and (**d**) CBD, ISS, and cisplatin in CRC cell lines based on cell viability. (**a**) Normalized isobologram for the combination of CBD and cisplatin with normalization of the dose with IC50 to the unity of both *x*- and *y*-axis in HCT-116, HT-29, and LS-174T. All the combination points indicated the antagonistic interaction. Abbreviations: A—cisplatin (D)1/(IC50)1; B—CBD (D)2/(IC50)2; D—dose; Point 1—the combination of CBD IC70 and ISS IC50; Point 2—the combination of CBD IC75 and cisplatin IC75. (**b**) Normalized isobologram for the combination of ISS and cisplatin with normalization of the dose with IC50 to the unity of both *x-* and *y*-axis in the HCT-116, HT-29, and LS-174T. Most of the combination points indicated synergistic interaction, except for the HCT-116 CRC cell line. Abbreviations: A—ISS (D)1/(IC50)1; B—cisplatin (D)2/(IC50)2; D—dose; Point 1—the combination of cisplatin IC70 and ISS IC50; Point 2—the combination of cisplatin IC50 and ISS IC50; Point 3—the combination of cisplatin IC20 and ISS IC50. (**c**) Normalized isobologram for the combination of ISS and CBD with normalization of the dose with IC50 to the unity of both *x*- and *y*-axis in HCT-116, HT-29, and LS-174T. Most of the combination points indicated synergistic interaction. Abbreviations: A—ISS (D)1/(IC50)1; B—CBD (D)2/(IC50)2; D—dose; Point 1—the combination of CBD IC70 and ISS IC50; Point 2—the combination of CBD IC50 and ISS IC50; Point 3—the combination of CBD IC20 and ISS IC50. (**d**) Fa–CI plot for the combination of ISS, CBD, and cisplatin in HCT-116, HT-29, and LS-174T. Abbreviations: CI—combination index; Fa—fraction affected by the drug concentration (% of cell growth inhibition/100); ISS + c—the combination of ISS with cisplatin in different doses; ISS + C—the combination of ISS with CBD in different doses; I + C+c—the combination of ISS with CBD and cisplatin in different doses. The CI was calculated to quantify synergism and antagonism of the drugs, where CI < 1 indicates synergism, CI = 1—additive effect, and CI > 1—antagonistic effect. Normalized isobologram and Fa–CI plots were generated using CompuSyn software.

**Figure 3 ijms-24-14743-f003:**
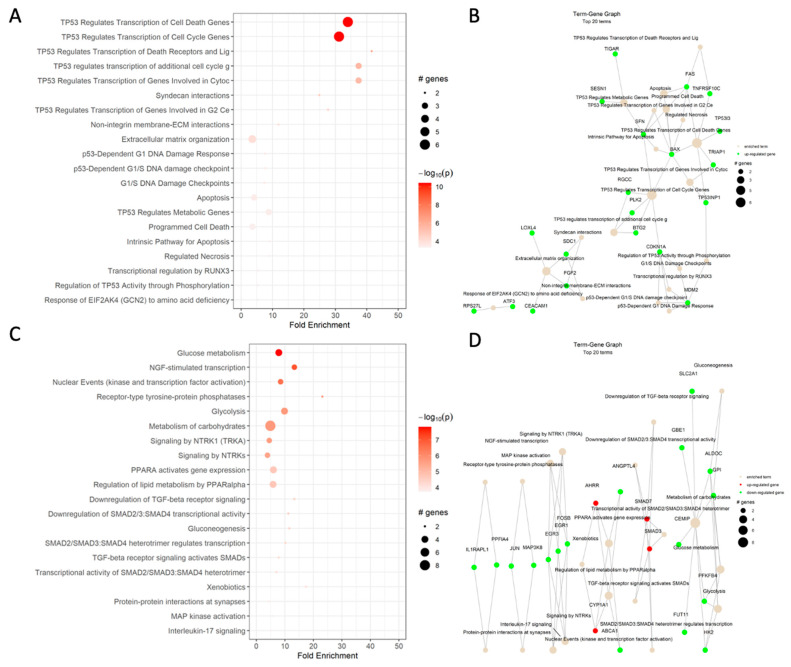
Reactome dot plot (left) and Term-Gene Graph of top 20 terms for (**A**,**B**) cisplatin vs. DMSO, (**C**,**D**) CBD vs. DMSO, (**E**,**F**) ISS vs. untreated, (**G**,**H**) cisplatin and CBD vs. DMSO, (**I**,**J**) cisplatin and ISS vs. DMSO, (**K**,**L**) CBD and ISS vs. DMSO, and (**M**,**N**) cisplatin, CBD, and ISS vs. DMSO in the HCT-116 CRC cell line. The top 20 terms with the highest fold change are shown in plots to the left, with their corresponding Reactome pathways on the *y*-axis. The size of the dots represents the number of genes in each pathway, while the red color’s intensity indicates the enrichment’s significance. The most enriched pathways are shown at the top of the plot. The graphs to the right side display the top 20 Reactome terms and their corresponding gene sets. Each node represents a Reactome term, and the size of the node is proportional to the number of genes associated with that term. The edges between the nodes represent the overlap in gene sets between the terms. The edge’s thickness represents the overlap’s magnitude, and the edge’s color represents the direction of the overlap, with green for upregulated and red for downregulated genes. Figures were generated using pathfindR enrichment analysis software.

**Table 1 ijms-24-14743-t001:** The IC50 values for CBD, ISS, and cisplatin in HCT-116, HT-29, and LS-174T CRC cell lines used in the experiments.

Treatment	HCT-116 IC50	HT-29 IC50	LS-174T IC50
Cisplatin	0.83 µM	1.17 µM	2.87 µM
CBD	5.56 µM	3.31 µM	4.80 µM
ISS *	69.99% (16 h)	69.78% (16 h)	72.43% (17 h)

* All cell lines were starved for 16 h out of 24 over five days of treatment, which corresponds to 66.7% of the time.

**Table 2 ijms-24-14743-t002:** **Summary of the selected terms affected by ISS, CBD, and cisplatin treatments in the HCT-116 CRC cell line.** Data are based on mRNA expression data as analyzed by Reactome analysis with PathfindR. All names of terms are as per PathfindR. ISS—intermittent serum starvation; DMSO—dissolves cisplatin and allows it to enter the cell; thus, the control group contained DMSO alone in order to analyze the effect of cisplatin on cells.

ISS vs. Untreated		CBD vs. DMSO		Cisplatin vs. DMSO	
Term	Fold Enrichment	Term	Fold Enrichment	Term	Fold Enrichment
ERBB2 activates PTK6 signaling	9.8	Receptor-type tyrosine-protein phosphatases	23.1	TP53 regulates transcription of cell death genes	41.5
PI3K events in ERBB4 signaling	9.1	Xenobiotics	17.3	TP53 regulates transcription of cell cycle genes	37.4
NGF-stimulated transcription	8.4	Regulation of gene expression by hypoxia-inducible factor	17.3	Syndecan interactions	37.4
ERBB2 regulates cell motility	7.3	Activation of the AP-1 family of transcription factors	17.3	TP53 regulates transcription of genes involved in G2 cell cycle arrest	34.0
Estrogen-dependent nuclear events downstream of ESR-membrane signaling	6.3	TP53 regulates transcription of death receptors and ligands	14.5	Intrinsic pathway of apoptosis	31.1
RAF-independent MAPK1/3 activation	5.5	Downregulation of TGF-ß signaling	13.3	Response of EIF2AK4 (GCN2) to amino acid deficiency	27.7
MAPK targets/nuclear events mediated by MAP kinases	4.2	FOXO-mediated transcription of cell cycle genes	10.2	TP53 regulates metabolic genes	24.9
Signaling by EGFR in cancer	3.8	Downregulation of TGF-ß signaling	13.3	Cellular response to starvation	24.9
NOTCH3 intracellular domain regulates transcription	3.8	Glucose metabolism	7.9	FOXO-mediated transcription	24.9
Interferon *α*/*β* signaling	3.4	PPAR*α* activates gene expression	6.0	POU5F1 (OCT4), SOX2, and NANOG activate genes related to proliferation	10.4

**Table 3 ijms-24-14743-t003:** **Summary of the selected terms affected by various ISS, CBD, and cisplatin combinations in the HCT-116 CRC cell line.** Data are based on mRNA expression data as analyzed with Reactome analysis using PathfindR. All names of terms are as per PathfindR.

Cisplatin + CBD vs. DMSO		Cisplatin + ISS vs. DMSO		CBD + ISS vs. DMSO		Cisplatin + CBD + ISS vs. DMSO	
Term	Fold Enrichment	Term	Fold Enrichment	Term	Fold Enrichment	Term	Fold Enrichment
Activation of the AP-1 family of transcription factors	15.9	NGF-stimulated transcription	9.5	ERBB2 activates PTK6 signaling	5.1	Condensation of prometaphase chromosomes	6.6
Aberrant regulation of mitotic G1/S transition in cancer due to Rb1 defects	14.1	PI3K events in ERBB4 signaling	8.2	Gluconeogenesis	4.4	Kinesins	4.4
Defective binding of RB1 mutants to E2F1, E2F2, and E2F3	14.1	RAF-independent MAPK1/3 activation	6.3	NFkB is activated and signals survival	4.0	Gluconeogenesis	4.0
FGFR3 mutant receptor activation	13.3	Estrogen-dependent nuclear events downstream of ESR-membrane signaling	6.3	PI3K cascade: FGFR4	3.3	NGF-stimulated transcription	4.0
TP53 regulates transcription of cell death genes	10.9	ERKs are inactivated	4.7	Glycolysis	3.0	RAF-independent MAPK1/3 activation	3.7
NGF-stimulated transcription	10.2	Activation of the AP-1 family of transcription factors	4.1	EGFR downregulation	2.5	PI3K cascade: FGFR4	3.6
MAPK targets/nuclear events mediated by MAP kinases	8.0	IL-17 signaling	2.4	Glucose metabolism	2.5	Resolution of sister chromatid cohesion	3.6
Insulin receptor signaling cascade	4.5	Interferon *α*/*β* signaling	2.3	Constitutive signaling by aberrant PI3K in cancer	2.4	Cyclin A/B1/B2-associated events during G2/M transition	3.4
TP53 regulates transcription of DNA repair genes	3.9	TP53 regulates transcription of cell death genes	2.3	TP53 regulates transcription of cell death genes	2.4	TP53 regulates transcription of genes involved in G2 cell cycle arrest	3.3
Glucose metabolism	2.7	PI3K/AKT signaling in cancer	2.0	Negative regulation of the PI3K/AKT network	2.2	TP53 regulates transcription of cell death genes	3.3

## Data Availability

mRNA sequencing data were uploaded to a public repository.

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
