# Peer review of "Transcriptome Analysis of Cisplatin, Cannabidiol, and Intermittent Serum Starvation Alone and in Various Combinations on Colorectal Cancer Cells"

_ijms, 2023, doi:10.3390/ijms241914743_

Round 1
Reviewer 1 Report
The manuscript by Viktoriia Cherkasova and co-authors provides a deep transcriptomic analysis of effects caused by cisplatin, cannabidiol with starvation. The experimental model was simple HCT-116 colorectal cancer cell line, however some preliminary experiments were conducted with two more CRC cell lines. The presented complex synergism between used treatment approach is impressing and I believe it might change the attitude to therapy of cancer patients.
The text is written with very good English, the complex results are explained in readable and clear way. The Authors sometimes use too long sentences. Since results are rich in details, the length of sentences may make them confusing or readers. Sometimes the main message is lost. For instance, sentences from lines 411-414.
I found only few aspects which could be improved. Detailed remarks are listed below:
1. Line 51: „CBD stimulates transient ion receptor ion channels (TRPV1, TRVP2)” – confusing phrase, consider rewriting.
2. Line 81 – What does the ISS abbreviation mean? All abbreviations usually should be explained at first place that appear in the text.
3. In my opinion, the Authors should add the information concerning the scientific aim of DMSO using in their experiments. That seems to be obvious, however, not all readers will be experienced enough to know it.
4. I also do not understand why HCT-116 cell line was chosen for extended experiments.
Taking into consideration that Authors took a lot of hard work to conduct all experiments and write a manuscript, it can be considered for publication in IJMS after minor improvement.
Author Response
Reviewer #1
The manuscript by Viktoriia Cherkasova and co-authors provides a deep transcriptomic analysis of effects caused by cisplatin, cannabidiol with starvation. The experimental model was simple HCT-116 colorectal cancer cell line, however some preliminary experiments were conducted with two more CRC cell lines. The presented complex synergism between used treatment approach is impressing and I believe it might change the attitude to therapy of cancer patients.
The text is written with very good English, the complex results are explained in readable and clear way. The Authors sometimes use too long sentences. Since results are rich in details, the length of sentences may make them confusing or readers. Sometimes the main message is lost. For instance, sentences from lines 411-414.
Our response
We corrected this sentence and also additionally proofread the entire manuscript for clarity.
I found only few aspects which could be improved. Detailed remarks are listed below:
- Line 51: „CBD stimulates transient ion receptor ion channels (TRPV1, TRVP2)” – confusing phrase, consider rewriting.
Our response: To improve clarity, we changed “stimulates” to “activates”.
- Line 81 – What does the ISS abbreviation mean? All abbreviations usually should be explained at first place that appear in the text.
Our response: It is intermittent serum starvation – this is now added in the text.
- In my opinion, the Authors should add the information concerning the scientific aim of DMSO using in their experiments. That seems to be obvious, however, not all readers will be experienced enough to know it.
Our response: We added information on DMSO in the Table description. “DMSO – is used to dissolve cisplatin and allow it to enter the cell, thus the control group should contain DMSO alone in order to analyze the effect of cisplatin on cells.”
- I also do not understand why HCT-116 cell line was chosen for extended experiments.
Our response: This cell line was used because the cisplatin IC50 level was the lowest in this cell line, compared to other cell lines; the use of lower amount of cisplatin would allow less “noise” at the level of mRNA in the treated cells. We added this to the text.
Reviewer 2 Report
Congratulations to Authors. This article is very well written. As mentioned, this project aimed to discover a novel treatment for CRC using drug combinations. This is very important milestone in cancer therapy and drug development. Your results provided evidence that combination of ISS and CBD suppresses cell metabolic pathways in HCT-116 CRC cell lines hence effecting cytotoxicity of cisplatin treatment.
There are some minor corrections to be considered.
Line 165 - 167:
To establish IC50s, a range of concentrations (1-15 μM) of cisplatin were obtained by diluting the cannabinoids in the fresh complete media and tested on CRC and normal colon epithelial cell line.
The highlighted area should be rewrite as diluting the cisplatin in fresh complete media ...
Line 184 - 185:
Cell viability was determined by MTT [3-(4, 5-dimethylthiazol-2-Yl)-2, 5-diphenylte-trazolium bromide] assay.
Yl should be corrected to yl.
Line 194 - 195:
Cells were then maintained at 37oC in a humidified atmosphere containing 5% CO2 for 24 hours.
37oC should be corrected to 37oC.
Line 206 - 208:
Once grown to 85-90% confluency in a Petri dish, cells were detached using TRYP- SIN/EDTA (0.25% Trypsin and 2.21 mM EDTA-4Na, Cat#325-043-EL, WISENT INC., Que- bec, Canada) and re-plated in 6-well plates at a density of 6 x 10 5 cells/well.
6 x 10 5 should be written as 6 x 105.
Line 217 - 218:
Additionally, the RNA integrity of RNA samples was assessed by agarose gel electrophoresis.
Author Response
There are some minor corrections to be considered.
Line 165 - 167:
To establish IC50s, a range of concentrations (1-15 μM) of cisplatin were obtained by diluting the cannabinoids in the fresh complete media and tested on CRC and normal colon epithelial cell line.
The highlighted area should be rewrite as diluting the cisplatin in fresh complete media ...
Our response: Noted and corrected.
Line 184 - 185:
Cell viability was determined by MTT [3-(4, 5-dimethylthiazol-2-Yl)-2, 5-diphenylte-trazolium bromide] assay.
Yl should be corrected to yl.
Our response: Corrected
Line 194 - 195:
Cells were then maintained at 37oC in a humidified atmosphere containing 5% CO2 for 24 hours.
37oC should be corrected to 37oC.
Our response: Corrected
Line 206 - 208:
Once grown to 85-90% confluency in a Petri dish, cells were detached using TRYP- SIN/EDTA (0.25% Trypsin and 2.21 mM EDTA-4Na, Cat#325-043-EL, WISENT INC., Que- bec, Canada) and re-plated in 6-well plates at a density of 6 x 10 5 cells/well.
6 x 10 5 should be written as 6 x 105.
Our response: Corrected
Line 217 - 218:
Additionally, the RNA integrity of RNA samples was assessed by agarose gel electrophoresis.
Our response: Corrected.